# Massive Spontaneous Pneumomediastinum—A Form of Presentation for Severe COVID-19 Pneumonia

**DOI:** 10.3390/medicina58111525

**Published:** 2022-10-26

**Authors:** Camelia Corina Pescaru, Monica Steluța Marc, Emanuela Oana Costin, Andrei Pescaru, Ana-Adriana Trusculescu, Adelina Maritescu, Noemi Suppini, Cristian Iulian Oancea

**Affiliations:** 1Pulmonology Department, ‘Victor Babes’ University of Medicine and Pharmacy, 300041 Timisoara, Romania; 2Center for Research and Innovation in Precision Medicine of Respiratory Diseases (CRIPMRD), ‘Victor Babes’ University of Medicine and Pharmacy, 300041 Timisoara, Romania; 3‘Victor Babes’ University of Medicine and Pharmacy, 300041 Timisoara, Romania

**Keywords:** COVID-19, SARS-CoV-2, spontaneous pneumomediastinum, subcutaneous emphysema

## Abstract

For COVID-19 pneumonia, many manifestations such as fever, dyspnea, dry cough, anosmia and tiredness have been described, but differences have been observed from person to person according to age, pulmonary function, damage and severity. In clinical practice, it has been found that patients with severe forms of infection with COVID-19 develop serious complications, including pneumomediastinum. Although two years have passed since the beginning of the pandemic with the SARS-CoV-2 virus and progress has been made in understanding the pathophysiological mechanisms underlying the COVID-19 infection, there are also unknown factors that contribute to the evolution of the disease and can lead to the emergence some complications. In this case report, we present a patient with COVID-19 infection who developed a massive spontaneous pneumomediastinum and subcutaneous emphysema during hospitalization, with no pre-existing lung pathology and no history of smoking. The patient did not get mechanical ventilation or chest trauma, but the possible cause could be severe alveolar inflammation. The CT results highlighted pneumonia in context with SARS-CoV-2 infection affecting about 50% of the pulmonary area. During hospitalization, lung lesions evolved 80% pulmonary damage associated with pneumomediastinum and subcutaneous emphysema. After three months, the patient completely recovered and the pneumomediastinum fully recovered with the complete disappearance of the lesions. Pneumomediastinum is a severe and rare complication in COVID-19 pneumonia, especially in male patients, without risk factors, and an early diagnosis can increase the chances of survival.

## 1. Introduction

Pneumomediastinum is a rare condition that represents the presence of air in the mediastinum. It is divided into two distinct types: spontaneous or secondary. Spontaneous pneumomediastinum is usually a self-condition, but it may be caused by precipitating factors such as: increased effort, recreational drugs, cough, vomiting effort, labor or Valsalva maneuvers [1].

Secondary pneumomediastinum is caused by a traumatic or damage to the mediastinum. It usually refers to an external factor such as chest injury, surgical complications, chronic lung disease, barotrauma or mechanical ventilation [2].

This condition generally occurs in young adults, especially in men [1]. The increased frequency in youth is due to the fact that their mediastinal tissue is loose, compared to the elders, whose tissues become fibrous with age. Thus, air can penetrate loose tissue much more easily than fibrous tissue [3].

Pneumomediastinum is a rare complication of COVID-19 pneumonia with an unknown mechanism, which is probably related to the increased alveolar pressure and pulmonary damage in people who develop severe forms of coronavirus disease [4]. This condition is based on the Macklin effect, which represents the alveolar rupture and air passage through the mediastinum due to increased thoracic pressure and severe inflammation [5].

The incidence of pneumomediastinum in COVID-19 pneumonia in patients without lung history disease is around 16.6%, as proved by an Iranian study [6].

Although it is considered a rare phenomenon, the prevalence of pneumomediastinum in COVID-19 pneumomia has been continually increasing in comparison with patients with adult respiratory distress syndrome since 2003, when it was 4%. This could be happened because of the compressed air injury, level of barotrauma and higher susceptibility in the population infected with SARS-CoV-2; it was also observed that men are more likely to develop the problem [7].

The novelty of this case report consists in the fact that a young patient without comorbidities except OSAS developed massive pneumomediastinum, including subcutaneous emphysema, for which the patient was monitored long-term and completely recovered 3 months after the onset.

In the clinical trial, the most common symptoms were chest pain with irradiation to the neck or back, unexplained and unexpected dyspnea and subcutaneous emphysema. It may include other symptoms such as cough, neck pain, or vomiting, or it could be asymptomatic. It is a life-threatening condition that must be carefully monitored [8,9].

## 2. Materials and Methods

### Case Presentation

A 45-year-old man with a history of essential hypertension and sleep apnea, both under treatment, came to the hospital presenting the following symptoms: a high-grade fever of 39 °C, shortness of breath, muscle pain, fatigue and anosmia. Blood pressure was 145/82 mmHg, the pulse was 95 bpm, his respiratory rate was 35 and oxygen saturation was 91% on room air. The real-time reverse transcription-polymerase chain reaction (RT-PCR) for SARS-CoV-2 was positive, but the patient refused hospitalization and went home with a treatment scheme based on symptomatics, vitamins and antibiotics. After 3 days, his grade of fever increased, his condition had depreciated and he decided to come back for investigations. Laboratory tests showed that the level of C-reactive protein, ferritin, LDH, D-dimers, were elevated, and the patient also had moderate hepatocytolysis and a high level of blood glucose (Table 1).

The initial CT results presented the appearance of pneumonia in the context of SARS-CoV-2 infection (moderate form), affecting approximately 50% of the lung area. He started treatment with antivirals in the first stage; corticosteroids, vitamins and anticoagulants according to guidelines. The patient benefited from antiviral therapy with Favipiravir—200 mg/tablet; on the first day 16 tablets (8–0–8), then 8 tablets/day (4–0–4) for 7 days; Anakinra—150 mg/mL, 4 syringes/ on the first day (2–0–2) then 1/day; Corticotherapy—Dexamethasone vial of 8 mg, 2 × 1/ day for 10 days; anticoagulant Fraxiparine 8600 iu/0.8 mL, once a day throughout the hospitalization period for prophylactic purposes; Pantoprazole 40 mg bottle, 2/day, for 10 days; ascorbic acid vial of 750 mg, 2/day, codeine 15 mg, 2 tablets/day; and oxygenoterapy 10 L/min through the reservoir mask, maintaining oxygen saturation over 94%. He remained on oxygen therapy, vitamins and antitussives for the entire hospitalization period.

The patient presented severe OSAS, but with hemodynamic stability and relatively well-tolerated daytime and nighttime symptoms, which correlated with the patient’s current situation, required the timing of the administration of nighttime treatment with CPAP-type continuous positive air pressure. This timing was taken into account precisely to prevent this air leak syndrome (pneumothorax with bronchopleural fistula). Compliance with hygiene–dietary rules with weight loss and nutritional counseling plus compliance with sleep hygiene rules were recommended.

During the hospitalization, on day 14, the patient presented coughing, dyspnea, retrosternal pain and crackles were noted around his neck and chest area; symptoms suggestive of pneumomediastinum and subcutaneous emphysema. A CT was performed and confirmed severe pneumomediastinum with extensive subcutaneous emphysema (Figure 1); the lung damage had increased to 80%. The thoracic surgeon team considered it better to wait before a surgical approach, mainly because a spontaneous pneumomediastinum is very probable to absorb by itself. After three days of continuous monitoring, the status of the patient gradually improved. He was included in respiratory rehabilitation programs due to his fatigue and difficulty in breathing, and the clinical status of the patient was improved. The subcutaneous emphysema disappeared with superficial palpation and his oxygen saturation improved to 95% on room air. Because of the respiratory rehabilitation exercises, the patient overcame his fear of breathing again and of making a minimum breathing effort (things he forgot during the illness). This improvement helped to reduce his anxiety and mental state. His recovery was almost completed, being discharged from hospital after 30 days in a stable condition. Three months later, the reassessment CT showed that pneumomediastinum and emphysema were completely absorbed (Figure 1).

## 3. Discussion

The development of spontaneous pneumomediastinum is a challenging complication for healthcare workers in patients with SARS-CoV2 infection. It can be spontaneous or secondary and represents a gaseous infiltration into mediastinal cellular tissues.

Although pneumomediastinum occurs most recently in COVID-19 pneumonia, the main causes of its occurrence remain the ones mentioned above. The factors that may increase the risk of developing this condition are smoking, male sex, age, asthma, COPD or symptoms such as prolonged cough and excessive vomiting. Subcutaneous crepitations occur when air gets into the tissues under the skin and represents the appearance of subcutaneous emphysema, a major sign of pneumomediastinum [10]. Radiology is fundamental in diagnostic pneumomediastin, *X*-ray being the first intent investigation. However, thorax CT is used to diagnose pneumomediastinum in situations where *X*-ray is not sufficient, and can provide additional information about pre-existing parenchymal or pleural pulmonary pathologies. Guidelines recommend having an *X*-ray for young patients who present unexplained dyspnea and chest pain [11].

We concluded that our patient developed spontaneous pneumomediastinum because of the severe injuries caused by COVID-19 infection, considering the fact that he did not have risk factors. The most probable mechanism of producing it was based on the Macklin effect, the alveolar rupture and air passage through the mediastinum, due to increased intrathoracic pressure and important inflammation [8]. The patient suddenly presented dyspnoea, chest pain, hypoxia and tachycardia, so a native thoracic CT scan was performed in order to exclude other differential diagnoses, such as pulmonary embolism. The normal value of D-dimers biologically ruled out the diagnosis of pulmonary thromboembolism in the dynamics throughout the hospitalization, and pneumothorax was ruled out by pulmonary CT.

The CT scan confirmed a huge pneumomediastinum, which challenged the medical team to consider therapeutic management for this interesting case, with such a complication of SARS-CoV-2 infection. Although the diagnosis was established, it was better to wait than take action. Therefore, in this case, it was preferred to provide vital functions and simple therapeutic measures, such as oxygen and analgesics, with a good outcome. Due to prompt radiological investigations in our hospital, patients with COVID-19 benefited from rapid diagnosis and correct evaluation of lung lesions, thus being able to provide adequate treatment and prompt management. This case was a successful one; the patient had a favorable evolution so that he was discharged with a good general condition, and at the control thoracic CT scan at 3 months, both the lung lesions and the pneumomediastinum were returned.

Although two years have passed since the beginning of the pandemic with the SARS-CoV-2 virus and progress has been made in understanding the pathophysiological mechanisms underlying the COVID-19 infection, there are also unknown factors that contribute to the evolution of the disease and can lead to the emergence of severe complications [12].

Anxiety is often present, but in patients who develop pneumomediastinum as a complication, the level of anxiety is increased and requires treatment [1]. Physical examination reveals crepitus at palpation when there is subcutaneous emphysema involved [13]. A sound like bursting balloons is frequently detected by the patient. The Hamman sign is pathognomic for mediastinal emphysema and represents a synchronous sound with the heartbeat, specifically produced by the difference of the heartbeat against air-filled tissues in the left precordial border [14]. The standard diagnosis is made by chest radiography or chest CT when an X-ray is inconclusive. The differential diagnosis should be made with conditions that present symptoms such as dyspnea, chest pain and hypoxia, so taking into consideration pulmonary embolism, pneumothorax, coronary syndrome or pericarditis is always necessary. Complications are rare, but if hypertensive pneumomendiastinum is involved, vessel compression could affect the venous blood return and this may compromise the hemodynamic and respiratory system. On the other hand, mediastinitis are also a serious complication, but the mortality, in this case, is increased by the coexisting illnesses [1]. Developing pneumomediastinum in this viral infection may be an indicator of worsening disease, but our patient luckily survived due to the proper investigations, diagnosis and suitable management, and his status was completely recovered [15].

## 4. Conclusions

In conclusion, pneumomediastinum in COVID-19 can also occur in young patients without pre-existing lung pathology with the exception of OSAS in this case but with acute respiratory distress syndrome which may represent the physiopathological production mechanism for pneumoemdiastinum. Although pneumomediastinum is a very rare complication in the evolution of the COVID-19 infection, it can be life-threatening for the patient. In the presented case, the evolution was favorable, with complete resorption of the pneumomediastinum within 3 months of the occurrence.

## Figures and Tables

**Figure 1 medicina-58-01525-f001:**
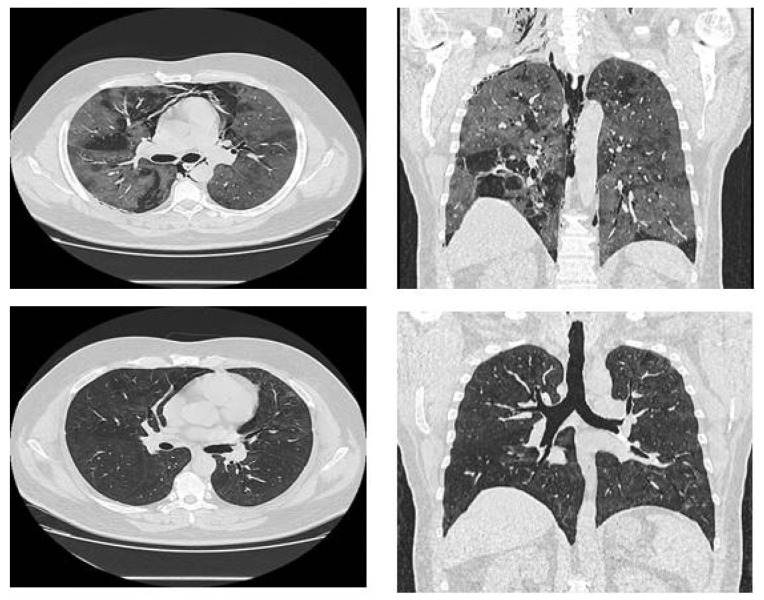
Thoracic CT showing pneumomediastinum and CT, three-months later, showing complete remission of pneumomediastinum.

**Table 1 medicina-58-01525-t001:** Complete blood count.

Laboratory Test	Conventional Units	Value	Reference Range Value
WBC	* 10^3^/µL	5.05	4.00–10.00
Lymphocytes	* 10^3^/µL	0.74	1.20–4.40
Monocyte	* 10^3^/µL	0.6	0.22–1.00
Interleukin 6	Pg/mL	50.75	<9.7
LDH	U/L	778	135–225
Blood glucose	mg/dL	126	74–106
Ferritin	µg/L	2861.6	30–400
D-dimers	µg/mL	0.62	<0.5
AST	U/L	226.1	0–40
ALT	U/L	124.8	0–41
CRP	mg/L	79.9	0–5
aPTT	seconds	25.4	25–36
PCR Covid	positive/negative	positive	-

* WBC, white blood cells; LDH, lactate dehydrogenase; AST, aspartate aminotransferase; ALT, alanine aminotransferase; CRP–C, reactive protein; aPTT, active partial thromboplastin time.

## Data Availability

The data presented in this study are available on reasonable request from the corresponding author.

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
