# Peer review of "Massive Spontaneous Pneumomediastinum—A Form of Presentation for Severe COVID-19 Pneumonia"

_medicina, 2022, doi:10.3390/medicina58111525_

Round 1

Reviewer 1 Report

Dear authors, the case report is interesting and valuable, and I would recommend publication although some corrections should be made first.

1.     The abstract is well structured, but I would suggest the substitution or removal of the word “severe” which appear five times.

2.     The introduction mentions similar cases of pneumomediastinum as well it discusses the physiopathology, but the text is too long. Some information could be better suitable to discussion such as the sixth and seventh paragraph (Line 68 until 90). Anxiety and pneumomediastinum should be discussed better in discussion topic.

3.     Line 55. This information about incidence should be included in the next paragraph with prevalence data.

4.     Line 73. “Anxiety is often present but in patients who develop pneumomediastinum as a complication, the level of anxiety is increased and requires treatment” please include the reference.

5.     Line 107. Oxygenoterapy 10L/min – please inform the supportive method offered (was it a reservoir mask? High-flow nasal cannula?).  Depending on the method use it may offer PEEP to generate or increases a pneumomediastinum.

6.     The discussion is well written but there is repetitive information: the mechanisms to form pneumomediastinum were already discussed in the introduction. The information appears again in Lines 136-139.

7.     A table including all reported cases of pneumomediastinum related to COVID-19 and its main characteristics should be interesting to understand the rarity about this complication.

8.     The conclusion involves again the pathological mechanisms. The conclusion should be about the case, and what we could learned about its presentation and evolution.

Author Response

The abstract is well structured, but I would suggest the substitution or removal of the word “severe” which appear five times.

Answer: Thank you for your recommendation. We modified the words accordingly.

The introduction mentions similar cases of pneumomediastinum as well it discusses the physiopathology, but the text is too long. Some information could be better suitable to discussion such as the sixth and seventh paragraph (Line 68 until 90). Anxiety and pneumomediastinum should be discussed better in discussion topic.

Answer: Thank you for your recommendation. We moved the paragraph as you suggested.

Line 55. This information about incidence should be included in the next paragraph with prevalence data.

Answer: Thank you for your recommendation. We moved the paragraph as you suggested.

Line 73. “Anxiety is often present but in patients who develop pneumomediastinum as a complication, the level of anxiety is increased and requires treatment” please include the reference.

Answer: Thank you for your recommendation. We introduced the reference.

Line 107. Oxygenoterapy 10L/min – please inform the supportive method offered (was it a reservoir mask? High-flow nasal cannula?).  Depending on the method use it may offer PEEP to generate or increases a pneumomediastinum.

Answer: Thank you for your recommendation. We described the medication, doses and duration of administration.

The discussion is well written but there is repetitive information: the mechanisms to form pneumomediastinum were already discussed in the introduction. The information appears again in Lines 136-139.

Answer: Thank you for your recommendation. We removed the sentence.

A table including all reported cases of pneumomediastinum related to COVID-19 and its main characteristics should be interesting to understand the rarity about this complication.

Answer: Thank you for your recommendation. Although a table would have been interesting to add, this was not the objective of our study. The allocated time to respond to the answers did not permit us to include such a table.

The conclusion involves again the pathological mechanisms. The conclusion should be about the case, and what we could learned about its presentation and evolution.

Answer: Thank you for your recommendation. We modified the conclusion accordingly.

Reviewer 2 Report

I would like to thank the respected editor of Medicina for the opportunity provided for me to review the manuscript entitled "Massive Spontaneous Pneumomediastinum – A form of Presentation for Severe COVID-19 Pneumonia". The investigators described a patient with COVID-19 and massive spontaneous pneumomediastinum successfully treated with conservative measures.

The case is interesting, the manuscript is well-written, and the conclusions are in line with the presented findings. However, I have several suggestions that (in my opinion) can improve the study's quality.

1.       Introduction: It is suggested to emphasize the novelty of this case report further over previously published ones (e.g., longer follow-up).

2.       Case presentation: It is suggested to describe this patient's sleep apnea treatment clearly. Continuous positive air pressure (CPAP), one of the most common treatment options for sleep apnea, can exert pulmonary barotrauma making the patient susceptible to air leakage syndrome.

https://doi.org/10.1155/2020/8898621

3.       Case presentation: It is suggested to clearly describe the treatment of the patient's COVID-19, including the drugs (antivirals and corticosteroids), their doses, and duration of administration. High doses of steroids can contribute to the weakening of the pulmonary interstitial tissue and subsequent air leakage.

https://doi.org/10.1016/j.radcr.2021.02.069

https://doi.org/10.1186/s13019-020-01308-7

4.       Case presentation: It is suggested to include the patient's 3-month follow-up CT scan indicating the resolution of the pneumomediastinum and emphysema.

5.       Discussion: It is suggested to further describe how other differential diagnoses (i.e., pulmonary embolism) were excluded.

Author Response

Introduction: It is suggested to emphasize the novelty of this case report further over previously published ones (e.g., longer follow-up).

Answer: Thank you for your recommendation. We modified the introduction part and emphasized the novelty of this case.

Case presentation: It is suggested to describe this patient's sleep apnea treatment clearly. Continuous positive air pressure (CPAP), one of the most common treatment options for sleep apnea, can exert pulmonary barotrauma making the patient susceptible to air leakage syndrome. https://doi.org/10.1155/2020/8898621

Answer: Thank you for your recommendation. We described the treatment for sleep apnea.

Case presentation: It is suggested to clearly describe the treatment of the patient's COVID-19, including the drugs (antivirals and corticosteroids), their doses, and duration of administration. High doses of steroids can contribute to the weakening of the pulmonary interstitial tissue and subsequent air leakage. https://doi.org/10.1016/j.radcr.2021.02.069 https://doi.org/10.1186/s13019-020-01308-7.

Answer: Thank you for your recommendation. We described the medication, doses and duration of administration.

Case presentation: It is suggested to include the patient's 3-month follow-up CT scan indicating the resolution of the pneumomediastinum and emphysema.

Answer: Thank you for your recommendation. CT at admission and CT at three months follow-up in figure 1.

Discussion: It is suggested to further describe how other differential diagnoses (i.e., pulmonary embolism) were excluded.

Answer: Thank you for your suggestion. The diagnosis of pulmonary thromboembolism was biologically ruled out by the normal value of D-dmers in the dynamics throughout the hospitalization and pneumothorax was ruled out by pulmonary CT. We added the sentence in the manuscript.

Round 2

Reviewer 1 Report

I recomenda the article acceptance.